# Semi-Supervised Partial Label Learning via Confidence-Rated Margin Maximization

**Wei Wang**    **Min-Ling Zhang**[*]
School of Computer Science and Engineering, Southeast University, Nanjing 210096, China
Key Laboratory of Computer Network and Information Integration
(Southeast University), Ministry of Education, China
`{wang_w, zhangml}@seu.edu.cn`

## Abstract

Partial label learning assumes inaccurate supervision where each training example is associated with a set of *candidate* labels, among which only one is valid. In many real-world scenarios, however, it is costly and time-consuming to assign candidate label sets to all the training examples. To circumvent this difficulty, the problem of *semi-supervised partial label learning* is investigated in this paper, where unlabeled data is utilized to facilitate model induction along with partial label training examples. Specifically, label propagation is adopted to instantiate the labeling confidence of partial label examples. After that, maximum margin formulation is introduced to jointly enable the induction of predictive model and the estimation of labeling confidence over unlabeled data. The derived formulation enforces confidence-rated margin maximization and confidence manifold preservation over partial label examples and unlabeled data. We show that the predictive model and labeling confidence can be solved via alternating optimization which admits QP solutions in either alternating step. Extensive experiments on synthetic as well as real-world data sets clearly validate the effectiveness of the proposed semi-supervised partial label learning approach.

## 1 Introduction

In partial label (PL) learning, each training example is represented by a single instance while associated with multiple candidate labels. It is assumed that the ground-truth label of PL training example resides in its candidate label set, which is not directly accessible to the training algorithm [8, 17, 32]. The need to learn from these inaccurate supervision information widely exists in various applications, such as image classification [6, 9, 30], ecoinformatics [4, 17, 33], web mining [18], natural language processing [23, 24, 35], etc.

Most partial label learning approaches work under supervised setting where the candidate labeling information is available for all training examples. In many real-world scenarios, however, the process of acquiring training examples with candidate labels might be demanding while abundant unlabeled data are readily available to facilitate model training. For instance, in crowdsourced image tagging, acquiring candidate annotations from web users for a large number of images would be costly and time-consuming while abundant unlabeled images can be easily collected from the web. Therefore, it is a natural remedy to consider semi-supervised partial label learning which exploits unlabeled data in conjunction with PL training examples to help induce predictive model with strong generalization performance.

---

[*]Corresponding author

Correspondingly, a novel approach named PARM, i.e. *semi-supervised Partial label learning via confidence-rated mARgin Maximization*, is proposed in this paper. To make use of unlabeled data, PARM chooses to jointly estimate the labeling confidence over unlabeled data and induce the desired multi-class classification model. Specifically, PARM considers confidence-rated margin which is maximized by preserving labeling confidence manifold structure between PL training examples and unlabeled data. PARM tackles the resulting formuation based on alternating optimization, where the predictive model and labeling confidence are updated in either alternating step with QP solutions. Comparative studies on both synthetic and real-world data sets show that PARM achieves favorable performance against state-of-the-art approaches in exploiting unlabeled data for partial label learning.

To the best of our knowledge, SSPL [27] corresponds to the only prior work which considers utilizing unlabeled data for partial label learning. Specifically, SSPL adopts graph-based techniques to disambiguate the labeing information between PL training examples and unlabeled data via label propagation. Due to the transductive nature of graph-based techniques, the resulting algorithm won't generalize to make prediction on unseen instances. To account for this issue, $k$NN rule is further employed to enable prediction on unseen instances. Consequently, SSPL has to store all the disambiguated PL training examples as well as unlabeled data during testing phase, which makes SSPL less efficient in terms of storage overhead and prediction time. Due to the inductive nature of maximum margin approach, PARM is capable of making predictions on unseen examples without resorting to extra procedure.

The rest of this paper is organized as follows. Firstly, we briefly review related work on partial label learning. Secondly, technical details of the proposed approach are presented. Thirdly, experimental results of comparative studies are reported. Finally, we conclude this paper.

## 2   Related Work

Partial label learning corresponds to the weakly supervised learning problem with inaccurate labeling information [36], where the ground-truth label of each PL training example is concealed within its candidate label set and not directly accessible to the learning algorithm. To learn from PL training examples, a natural strategy is trying to disambiguate the candidate label set. One way to instantiate the disambiguation strategy is to treat the ground-truth label as latent variable, and then identify its value via iterative optimization procedure such as EM. Accordingly, the objective function for identification-based disambiguation can be defined based on the maximum likelihood criterion [15, 17, 19], maximum margin criterion [5, 22, 29], etc. Another way to instantiate the disambiguation strategy is to treat all candidate labels in an equal manner, and then make final prediction by averaging the modeling outputs from all candidate labels. Accordingly, the prediction rule for averaging-based disambiguation can be defined based on convex formulation [8], instance-based formulation [11, 14, 31], etc.

For the disambiguation strategy, one potential issue lies in that the effectiveness of disambiguation would be largely affected by the false positive labels within candidate label set. As the size of candidate label set increases, it is highly possible that the identified ground-truth label might turn out to be false positive one for identification-based disambiguation, while the modeling output from ground-truth label would be overwhelmed by those from false positive labels for averaging-based disambiguation. In light of this, another strategy to learn from PL training examples is trying to transform the partial label learning problem into other well-established learning problems. Accordingly, the transformation strategy can be instantiated based on binary decomposition [28, 32], dictionary learning [7], graph matching [20], regression [10, 26, 33], etc.

To help deal with the difficulty brought by weak supervision, one natural choice is to make use of the unlabeled data which are readily available for model training [36]. Semi-supervised learning [39] aims to make use of unlabeled data for training and typically learns from few labeled training examples together with large amount of unlabeled data. There are four major categories of semi-supervised learning methods, including graph-based approaches[2, 34, 38], disagreement-based approaches[3, 37], generative approaches [21] and low-density separation approaches[1, 16]. Graph-based approaches construct a graph and propagate label information to unlabeled data following the cluster assumption or manifold assumption. Disagreement-based approaches build multiple clssifiers and make use of the disagreement among them to facilitate the learning process. Generative approaches treat the missing labels as latent variables and estimate them via iterative process. Low-

density separation approaches often constrain the decision boundary to go across low-density regions in the feature space.

In the next section, a novel semi-supervised partial label learning approach is introduced to learn from PL training examples and unlabeled data in an inductive manner.

## 3 The Proposed Approach

Let $\mathcal{X} = \mathbb{R}^n$ denote the $n$-dimensional feature space and $\mathcal{Y} = \{y_1, y_2, \ldots, y_q\}$ denote the label space with $q$ class labels. Given the set of PL training examples $\mathcal{D}_P = \{(\boldsymbol{x}_i, S_i) \mid 1 \leq i \leq p\}$, where $\boldsymbol{x}_i \in \mathcal{X}$ is a $n$-dimensional feature vector $[x_{i1}, x_{i2}, \ldots, x_{in}]^{\mathrm{T}}$ and $S_i \subseteq \mathcal{Y}$ is the candidate label set associated with $\boldsymbol{x}_i$ . For each PL training example, it is assumed that the ground-truth label $y_i$ for $\boldsymbol{x}_i$ is concealed within its candidate label set $S_i$, i.e. $y_i \in S_i$. Furthermore, given the set of unlabeled data $\mathcal{D}_U = \{\boldsymbol{x}_i \mid p + 1 \leq i \leq p + u\}$, the task of semi-supervised partial label learning is to learn a multi-class classifier $f : \mathcal{X} \to \mathcal{Y}$ from $\mathcal{D}_P \bigcup \mathcal{D}_U$.

Let $\boldsymbol{f}_i = [f_{i1}, f_{i2}, \ldots, f_{iq}]^{\top}$ denote the labeling confidence vector for $\boldsymbol{x}_i$ $(1 \leq i \leq p + u)$ with $f_{il} \in [0, 1]$ and $\sum_{l=1}^q f_{il} = 1$. Correspondingly, we have the labeling confidence matrix $\mathbf{F}_P = [\boldsymbol{f}_1, \boldsymbol{f}_2, \ldots, \boldsymbol{f}_p]^{\top} \in [0, 1]^{p \times q}$ for PL training examples and $\mathbf{F}_U = [\boldsymbol{f}_{p+1}, \boldsymbol{f}_{p+2}, \ldots, \boldsymbol{f}_{p+u}]^{\top} \in [0, 1]^{u \times q}$ for unlabeled data. For ease of notations, we further define the feature mapping function $\Phi(\boldsymbol{x}, y) : \mathcal{X} \times \mathcal{Y} \mapsto \mathbb{R}^{nq}$ to be used in follow-up derivations:

$$\Phi(\boldsymbol{x}, y) = \begin{pmatrix} \boldsymbol{x} \cdot \mathbb{I}(y = y_1) \\ \boldsymbol{x} \cdot \mathbb{I}(y = y_2) \\ \cdots \\ \boldsymbol{x} \cdot \mathbb{I}(y = y_q) \end{pmatrix}. \tag{1}$$

Here, $\mathbb{I}(\pi)$ returns 1 if predicate $\pi$ holds. Otherwise, $\mathbb{I}(\pi)$ returns 0.

We choose to estimate the labeling confidence values for $\mathbf{F}_P$ via the label propagation procedure, which has been shown to be effective in disambiguating PL training examples [10, 25, 27, 31]. For each PL training example $(\boldsymbol{x}_i, S_i) \in \mathcal{D}_P$, let $\mathcal{N}_P(\boldsymbol{x}_i)$ be the set of $\boldsymbol{x}_i$'s $k$-nearest neighbours identified in $\mathcal{D}_P$. Then, the similarity matrix $\mathbf{W}_P = [w_{ij}^P]_{p \times p}$ over PL training examples is instantiated as: $w_{ij}^P = \exp\left(-\frac{\|\boldsymbol{x}_i - \boldsymbol{x}_j\|^2}{2\sigma^2}\right)$ if $\boldsymbol{x}_j \in \mathcal{N}_P(\boldsymbol{x}_i)$ and $w_{ij}^P = 0$ otherwise. We further normalize $\mathbf{W}_P$ by row to yield the propagation matrix $\mathbf{H} = \mathbf{D}_P^{-1} \mathbf{W}_P$ with $\mathbf{D}_P = \mathrm{diag}[d_1^P, d_2^P, \ldots, d_p^P]$ and $d_i^P = \sum_{j=1}^p w_{ij}^P$. Accordingly, the initial labeling confidence matrix $\mathbf{F}_P^{(0)}$ is set as:

$$\forall 1 \leq i \leq p : \quad f_{ij}^{(0)} = \begin{cases} \dfrac{1}{|S_i|}, & j \in S_i \\ 0, & \text{otherwise} \end{cases} \tag{2}$$

Thereafter, the following iterative label propagation procedure is invoked to update $\mathbf{F}_P$ until convergence:

$$\widetilde{\mathbf{F}}_P^{(t)} = \alpha \cdot \mathbf{H}\mathbf{F}_P^{(t-1)} + (1 - \alpha) \cdot \mathbf{F}_P^{(t-1)} \tag{3}$$

$$\forall 1 \leq i \leq p : \quad f_{il}^{(t)} = \begin{cases} \dfrac{\widetilde{f}_{il}^{(t)}}{\sum_{y_{l'} \in S_i} \widetilde{f}_{il'}^{(t)}}, & l \in S_i \\ 0, & \text{otherwise} \end{cases}$$

Here, $\alpha \in (0, 1)$ corresponds to the balancing parameter for label propagation.[2]

For each unlabeled data $\boldsymbol{x}_i \in \mathcal{D}_U$, let $\mathcal{N}_U(\boldsymbol{x}_i)$ be the set of $\boldsymbol{x}_i$'s $k$-nearest neighbours identified in $\mathcal{D}_P$. Similarly, we set the similarity matrix $\mathbf{W}_{UP} = [w_{ij}^{UP}]_{u \times p}$ between unlabeled data and PL training examples as: $w_{ij}^{UP} = \exp\left(-\frac{\|\boldsymbol{x}_{p+i} - \boldsymbol{x}_j\|^2}{2\sigma^2}\right)$ if $\boldsymbol{x}_j \in \mathcal{N}_U(\boldsymbol{x}_{p+i})$ and $w_{ij}^{UP} = 0$ otherwise. We also normalize $\mathbf{W}_{UP}$ by row to yield $\mathbf{S} = [s_{ij}]_{u \times p}$ such that $\mathbf{S} = \mathbf{D}_{UP}^{-1} \mathbf{W}_{UP}$ with $\mathbf{D}_{UP} = \mathrm{diag}[d_1^{UP}, d_2^{UP}, \ldots, d_u^{UP}]$ and $d_i^{UP} = \sum_{j=1}^p w_{ij}^{UP}$.

For the proposed PARM approach, the predictive model $\boldsymbol{w} \in \mathbb{R}^{nq}$ and the labeling confidence matrix $\mathbf{F}_U$ over unlabeled data are jointly optimized by solving the following *confidence-rated margin maximization* problem:

$$\min_{\boldsymbol{w},\boldsymbol{\Xi},\mathbf{F}_U} \quad \frac{1}{2}\|\boldsymbol{w}\|_2^2 + \frac{\lambda}{p}\sum_{i=1}^{p}\sum_{l=1}^{q} f_{il}\xi_{il} + \frac{\mu}{u}\sum_{i=p+1}^{p+u}\sum_{l=1}^{q} f_{il}\xi_{il} + \gamma\sum_{i=1}^{u}\sum_{j=1}^{p} s_{ij}\|\boldsymbol{f}_{p+i}-\boldsymbol{f}_j\|_2^2 \quad (4)$$

$$\text{s.t. } \boldsymbol{w}^{\mathrm{T}}\Phi(\boldsymbol{x}_i,y_l) - \max_{y_{l'}\neq y_l}\boldsymbol{w}^{\mathrm{T}}\Phi(\boldsymbol{x}_i,y_{l'}) \geq 1-\xi_{il}, \quad (1\leq i\leq p+u, \ 1\leq l\leq q)$$

$$\xi_{il}\geq 0, \quad (1\leq i\leq p+u, \ 1\leq l\leq q)$$

$$f_{il}\geq 0, \quad (p+1\leq i\leq p+u, \ 1\leq l\leq q)$$

$$\sum_{l=1}^{q} f_{il}=1, \quad (p+1\leq i\leq p+u)$$

Here, $\boldsymbol{\Xi} = [\xi_{il}]_{(p+u)\times q}$ corresponds to the set of slack variables with $\xi_{il}$ characterizing the multi-class classification margin. As shown in the second and third terms of the above objective function, $\xi_{il}$ is further rated by $f_{il}$ to account for the labeling confidence of $y_l$ being the ground-truth label for $\boldsymbol{x}_i$. To make full use of available supervision information, the estimated labeling confidence $\mathbf{F}_P$ over PL training examples are utilized in the fourth term to enforce manifold consistency between $\mathbf{F}_P$ and $\mathbf{F}_U$. To solve the derived problem, PARM employs alternating optimization to iteratively update $\boldsymbol{w}$ and $\mathbf{F}_U$.

**Fix $\boldsymbol{w}$, Optimize $\mathbf{F}_U$** When $\boldsymbol{w}$ is fixed, according to the first and second constraints in Eq.(4), we can have the values of slack variables as:

$$\xi_{il} = \max\left(0, 1+\max_{y_{l'}\neq y_l}\boldsymbol{w}^{\mathrm{T}}\Phi(\boldsymbol{x}_i,y_{l'}) - \boldsymbol{w}^{\mathrm{T}}\Phi(\boldsymbol{x}_i,y_l)\right) \quad (5)$$

Thereafter, the optimization problem in Eq.(4) turns out to be:

$$\min_{\mathbf{F}_U} \quad \frac{\mu}{u}\sum_{i=p+1}^{p+u}\sum_{l=1}^{q} f_{il}\xi_{il} + \gamma\sum_{i=1}^{u}\sum_{j=1}^{p} s_{ij}\|\boldsymbol{f}_{p+i}-\boldsymbol{f}_j\|_2^2 \quad (6)$$

$$\text{s.t. } f_{il}\geq 0, \quad (p+1\leq i\leq p+u, \ 1\leq l\leq q)$$

$$\sum_{l=1}^{q} f_{il}=1, \quad (p+1\leq i\leq p+u)$$

Note that Eq.(6) corresponds to a quadratic programming (QP) problem with $uq$ variables and $u(q+1)$ constraints, whose computational complexity would be demanding if $uq$ is large. To improve efficiency, we can decompose Eq.(6) into $u$ QP sub-problems each with $q$ variables and $q+1$ constraints. Without loss of generality, the labeling confidence vector $\boldsymbol{f}_i$ for unlabeled data $\boldsymbol{x}_i$ $(p+1\leq i\leq p+u)$ can be optimized by fixing the values of other elements in $\mathbf{F}_U$:

$$\min_{\boldsymbol{f}_i} \quad \gamma\boldsymbol{f}_i^{\mathrm{T}}\boldsymbol{f}_i + \left(\frac{\mu}{u}\boldsymbol{\xi}_i - 2\gamma\sum_{j=1}^{p} s_{ij}\boldsymbol{f}_j\right)^{\mathrm{T}}\boldsymbol{f}_i \quad (7)$$

$$\text{s.t. } f_{il}\geq 0, \quad (1\leq l\leq q)$$

$$\sum_{l=1}^{q} f_{il}=1$$

Here, $\boldsymbol{\xi}_i = [\xi_{i1}, \xi_{i2}, \ldots, \xi_{iq}]^{\top}$ is the slack vector for $\boldsymbol{x}_i$.

**Fix $\mathbf{F}_U$, Optimize $\boldsymbol{w}$** When $\mathbf{F}_U$ is fixed, the optimization problem in Eq.(4) turns out to be:

$$\min_{\boldsymbol{w},\boldsymbol{\Xi}} \quad \frac{1}{2}\|\boldsymbol{w}\|_2^2 + \frac{\lambda}{p}\sum_{i=1}^{p}\sum_{l=1}^{q} f_{il}\xi_{il} + \frac{\mu}{u}\sum_{i=p+1}^{p+u}\sum_{l=1}^{q} f_{il}\xi_{il} \quad (8)$$

$$\text{s.t. } \boldsymbol{w}^{\mathrm{T}}\Phi(\boldsymbol{x}_i,y_l) - \max_{y_{l'}\neq y_l}\boldsymbol{w}^{\mathrm{T}}\Phi(\boldsymbol{x}_i,y_{l'}) \geq 1-\xi_{il}, \quad (1\leq i\leq p+u, \ 1\leq l\leq q)$$

$$\xi_{il}\geq 0, \quad (1\leq i\leq p+u, \ 1\leq l\leq q)$$

Table 1: Pseudo-code of PARM.

**Inputs:**
$\mathcal{D}_P$:      the set of PL training examples $\{(\boldsymbol{x}_i, S_i) \mid 1 \le i \le p\}$
$\mathcal{D}_U$:      the set of unlabeled data $\{\boldsymbol{x}_i \mid p+1 \le i \le p+u\}$
$\lambda, \mu, \gamma$:      regularization parameters in Eq.(4)
$\boldsymbol{x}_*$:      unseen instance

**Outputs:**
$y_*$:      predicted class label for $\boldsymbol{x}_*$

**Process:**
1: Estimate the labeling confidence matrix $\mathbf{F}_P$ over PL training examples according to Eq.(3);
2: Initialize $\mathbf{F}_U$ by solving Eq.(6) with $\mu = 0$;
3: **repeat**
4:   Obtain $\boldsymbol{\alpha}$ by solving a series of QP subproblems in Eq.(12);
5:   Update $\boldsymbol{w}$ according to Eq.(13);
6:   Update $\mathbf{F}_U$ by solving a series of QP subproblems Eq.(7);
7: **until** convergence
8: Return $y_*$ according to Eq.(14).

For simplicity, the first and second constraints in Eq.(8) can be rewritten as: $\boldsymbol{w}^{\mathrm{T}}\Phi(\boldsymbol{x}_i, y_l) + \delta_{lr} - \boldsymbol{w}^{\mathrm{T}}\Phi(\boldsymbol{x}_i, y_r) \ge 1 - \xi_{il} \; (1 \le i \le p+u, \; 1 \le l, r \le q)$. Here, $\delta_{lr} = 1$ if $l = r$ and $\delta_{lr} = 0$ otherwise. Then, the Lagrangian of Eq.(8) corresponds to:

$$\mathcal{L}(\boldsymbol{w}, \boldsymbol{\Xi}, \boldsymbol{\alpha}) = \frac{1}{2}\|\boldsymbol{w}\|_2^2 + \frac{\lambda}{p}\sum_{i=1}^{p}\sum_{l=1}^{q} f_{il}\xi_{il} + \frac{\mu}{u}\sum_{i=p+1}^{p+u}\sum_{l=1}^{q} f_{il}\xi_{il} \qquad (9)$$

$$+ \sum_{i=1}^{p+u}\sum_{l=1}^{q}\sum_{r=1}^{q}\alpha_{lr}^i\left(\boldsymbol{w}^{\mathrm{T}}\Phi(\boldsymbol{x}_i, y_r) - \boldsymbol{w}^{\mathrm{T}}\Phi(\boldsymbol{x}_i, y_l) - \delta_{lr} + 1 - \xi_{il}\right)$$

where $\boldsymbol{\alpha} = [\alpha_{11}^1, \ldots, \alpha_{lr}^i, \ldots, \alpha_{qq}^{p+u}]^{\top}$ correspond to the Lagrangian multipliers with $\alpha_{lr}^i \ge 0$ ($1 \le i \le p+u, \; 1 \le l, r \le q$). By setting the gradient of $\mathcal{L}(\boldsymbol{w}, \boldsymbol{\Xi}, \boldsymbol{\alpha})$ w.r.t. $\boldsymbol{w}$ and $\boldsymbol{\Xi}$ to zero, we can have the dual problem of Eq.(8) as follows:

$$\min_{\boldsymbol{\alpha}} \; \frac{1}{2}\sum_{i=1}^{p+u}\sum_{j=1}^{p+u}\boldsymbol{x}_i^{\mathrm{T}}\boldsymbol{x}_j\sum_{l=1}^{q}\left(\sum_{r=1}^{q}\alpha_{lr}^i - \sum_{r=1}^{q}\alpha_{rl}^i\right)\left(\sum_{r=1}^{q}\alpha_{lr}^j - \sum_{r=1}^{q}\alpha_{rl}^j\right) + \sum_{i=1}^{p+u}\sum_{l=1}^{q}\sum_{r=1}^{q}\alpha_{lr}^i\delta_{lr} \quad (10)$$

$$\text{s.t. } \alpha_{lr}^i \ge 0, \quad (1 \le i \le p+u, \; 1 \le l, r \le q)$$

Note that Eq.(10) is a QP problem with $(p+u)q^2$ variables and $(p+u)q^2$ constraints, which would be difficult to be efficiently solved when $p+u$ or $q$ is large. Therefore, we decompose Eq.(10) into $p+u$ sub-problems each with $q^2$ variables and $q^2$ constraints. For ease of notations, we group the Lagrangian multipliers w.r.t. $\boldsymbol{x}_i$ into $\boldsymbol{\alpha}^i = [\alpha_{lr}^i]_{q \times q}$ and introduce the following terms $\mathbf{M} \in \{0,1\}^{q \times q^2}$ and $\mathbf{N} \in \{0,1\}^{q \times q^2}$:

$$\boldsymbol{\alpha}^i = \begin{bmatrix} \alpha_{11}^i & \alpha_{12}^i & \cdots & \alpha_{1q}^i \\ \alpha_{21}^i & \alpha_{22}^i & \cdots & \alpha_{2q}^i \\ \vdots & \vdots & \ddots & \vdots \\ \alpha_{q1}^i & \alpha_{q2}^i & \cdots & \alpha_{qq}^i \end{bmatrix}, \; \mathbf{M} = [\mathbf{I}_{q \times q}, \cdots, \mathbf{I}_{q \times q}], \; \mathbf{N} = \begin{bmatrix} \mathbf{1}_{1 \times q} & \mathbf{0}_{1 \times q} & \cdots & \mathbf{0}_{1 \times q} \\ \mathbf{0}_{1 \times q} & \mathbf{1}_{1 \times q} & \cdots & \mathbf{0}_{1 \times q} \\ \vdots & \vdots & \ddots & \vdots \\ \mathbf{0}_{1 \times q} & \mathbf{0}_{1 \times q} & \cdots & \mathbf{1}_{1 \times q} \end{bmatrix} \quad (11)$$

Here, $\mathbf{I}_{q \times q}$ is the identity matrix. Without loss of generality, $\boldsymbol{\alpha}^i$ can be optimized by fixing the values of other Lagrangian multipliers in $\boldsymbol{\alpha}$:

$$\min_{\boldsymbol{\alpha}^i} \; \frac{1}{2}\boldsymbol{x}_i^{\mathrm{T}}\boldsymbol{x}_i \mathrm{vec}(\boldsymbol{\alpha}^i)^{\mathrm{T}}\mathbf{C}^{\mathrm{T}}\mathbf{C}\mathrm{vec}(\boldsymbol{\alpha}^i) + \left(\sum_{j \ne i}\boldsymbol{x}_i^{\mathrm{T}}\boldsymbol{x}_j\mathbf{C}^{\mathrm{T}}\mathbf{C}\mathrm{vec}(\boldsymbol{\alpha}^j) + \mathrm{vec}(\mathbf{I}_{q \times q})\right)^{\mathrm{T}}\mathrm{vec}(\boldsymbol{\alpha}^i) \quad (12)$$

$$\text{s.t. } \alpha_{lr}^i \ge 0, \quad (1 \le l, r \le q)$$

Table 2: Characteristics of the experimental data sets.

| Controlled UCI Data Sets | | | | |
|---|---|---|---|---|
| **Data Set** | **# Examples** | **# Features** | **# Class Labels** | **# False Positive Labels** ($r$) |
| **Deter** | 358 | 23 | 6 | $r = 1, 2, 3$ |
| **Vehicle** | 846 | 18 | 4 | $r = 1, 2$ |
| **Abalone** | 4,177 | 7 | 29 | $r = 1, 2, 3$ |
| **Satimage** | 6,435 | 36 | 7 | $r = 1, 2, 3$ |

| Real-World Data Sets | | | | | |
|---|---|---|---|---|---|
| **Data Set** | **# Examples** | **# Features** | **# Class Labels** | **Avg. # CLs** | **Task Domain** |
| **Lost** | 1,122 | 108 | 16 | 2.23 | *automatic face naming* |
| **Mirflickr** | 2,780 | 1536 | 14 | 2.76 | *web image classification* |
| **BirdSong** | 4,998 | 38 | 13 | 2.18 | *bird song classification* |
| **LYN10** | 16,526 | 163 | 10 | 1.84 | *automatic face naming* |
| **LYN20** | 17,511 | 163 | 20 | 1.85 | *automatic face naming* |

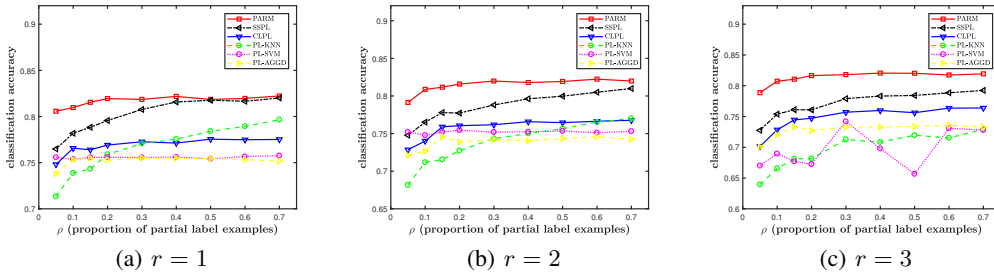

    (a) $r = 1$                     (b) $r = 2$                     (c) $r = 3$

Figure 1: Classification accuracy of each comparing approach changes as the proportion of PL training examples $\rho$ increases from 0.05 to 0.7 (Data set: `Satimage`; $r = 1, 2, 3$).

Here, $\text{vec}(\cdot)$ is the vectorization operator and $\mathbf{C} = \mathbf{M} - \mathbf{N}$.

As the alternating optimization procedure for $\boldsymbol{w}$ and $\mathbf{F}_U$ terminates, the predictive model $\boldsymbol{w} = [\boldsymbol{w}^1; \boldsymbol{w}^2; \ldots; \boldsymbol{w}^q]$ can be obtained based on the KKT condition:

$$\boldsymbol{w}_r = \sum_{i=1}^{p+u} \Big( \sum_{l=1}^{q} \alpha_{rl}^i - \sum_{l=1}^{q} \alpha_{lr}^i \Big) \boldsymbol{x}_i \quad (1 \leq r \leq q) \tag{13}$$

Accordingly, given the unseen instance $\boldsymbol{x}_*$, it is natural for PARM to predict its class label $y_*$ as:

$$y_* = \arg\max_{y \in \mathcal{Y}} \ \boldsymbol{w}^{\mathrm{T}} \Phi(\boldsymbol{x}_*, y) \tag{14}$$

In summary, Table 1 gives the pseudo-code of PARM. Firstly, the labeling confidence matrix $\mathbf{F}_P$ over PL training examples is estimated (Step 1). After that, an alternating optimization procedure is invoked to update predictive model $\boldsymbol{w}$ and the labeling confidence matrix $\mathbf{F}_U$ over unlabeled data (Steps 2-7). Finally, the class label for unseen instance is predicted based on the learned classification model (Step 8).

## 4 Experiments

### 4.1 Experimental Setup

The performance of PARM is compared against five state-of-the-art partial label learning algorithms, each configured with parameters suggested in respective literatures: 1) SSPL [27]: The only available semi-supervised partial label learning approach which learns from PL training examples and unlabeled data via graph-based label propagation [suggested configuration: $k = 10, \alpha = 0.7, \tilde{\beta} = 0.25, r =$

Table 3: Classification accuracy (mean±std) of each comparing approach on real-world partial label data sets (with $\rho \in \{0.05, 0.1, 0.15, 0.3, 0.5, 0.7\}$). In addition, ●/○ indicates whether PARM is statistically superior/inferior to the comparing approach on each data set (pairwise $t$-test at 0.05 significance level).

| Comparing | Lost | | | | | |
|---|---|---|---|---|---|---|
| approach | $\rho = 0.05$ | $\rho = 0.1$ | $\rho = 0.15$ | $\rho = 0.3$ | $\rho = 0.5$ | $\rho = 0.7$ |
| PARM | 0.267±0.052 | 0.339±0.079 | 0.422±0.044 | 0.571±0.057 | 0.647±0.053 | 0.701±0.035 |
| SSPL | 0.284±0.049 | 0.373±0.061 | 0.455±0.056 | 0.521±0.043● | 0.581±0.034● | 0.596±0.045● |
| PL-KNN | 0.188±0.039● | 0.253±0.039● | 0.261±0.028● | 0.332±0.039● | 0.410±0.046● | 0.445±0.019● |
| CLPL | 0.255±0.043 | 0.309±0.052 | 0.315±0.034● | 0.502±0.045● | 0.659±0.041 | 0.695±0.028 |
| PL-SVM | 0.118±0.043● | 0.221±0.082● | 0.287±0.080● | 0.482±0.069● | 0.580±0.070● | 0.681±0.048 |
| PL-AGGD | 0.290±0.041 | 0.185±0.042● | 0.334±0.031● | 0.586±0.044 | 0.635±0.035 | 0.672±0.043 |
| Comparing | Mirflickr | | | | | |
| approach | $\rho = 0.05$ | $\rho = 0.1$ | $\rho = 0.15$ | $\rho = 0.3$ | $\rho = 0.5$ | $\rho = 0.7$ |
| PARM | 0.436±0.020 | 0.487±0.029 | 0.560±0.033 | 0.578±0.039 | 0.614±0.037 | 0.633±0.032 |
| SSPL | 0.437±0.053 | 0.469±0.048 | 0.503±0.037● | 0.515±0.042● | 0.546±0.026● | 0.544±0.028● |
| PL-KNN | 0.389±0.048● | 0.443±0.034● | 0.465±0.034● | 0.495±0.027● | 0.511±0.024● | 0.536±0.040● |
| CLPL | 0.442±0.028 | 0.472±0.036 | 0.508±0.026● | 0.540±0.025● | 0.567±0.032● | 0.567±0.026● |
| PL-SVM | 0.129±0.063● | 0.214±0.079● | 0.292±0.108● | 0.352±0.100● | 0.449±0.107● | 0.492±0.062● |
| PL-AGGD | 0.477±0.039○ | 0.505±0.029○ | 0.524±0.045● | 0.536±0.041● | 0.518±0.026● | 0.519±0.035● |
| Comparing | BirdSong | | | | | |
| approach | $\rho = 0.05$ | $\rho = 0.1$ | $\rho = 0.15$ | $\rho = 0.3$ | $\rho = 0.5$ | $\rho = 0.7$ |
| PARM | 0.554±0.043 | 0.581±0.026 | 0.586±0.039 | 0.610±0.028 | 0.608±0.021 | 0.607±0.019 |
| SSPL | 0.457±0.025● | 0.504±0.022● | 0.529±0.025● | 0.563±0.030● | 0.588±0.033 | 0.597±0.027 |
| PL-KNN | 0.405±0.023● | 0.443±0.025● | 0.465±0.024● | 0.508±0.025● | 0.527±0.028● | 0.537±0.020● |
| CLPL | 0.525±0.026 | 0.536±0.021● | 0.566±0.020● | 0.603±0.021 | 0.612±0.019 | 0.618±0.019 |
| PL-SVM | 0.538±0.043 | 0.589±0.046 | 0.602±0.024 | 0.588±0.031 | 0.609±0.028 | 0.597±0.022 |
| PL-AGGD | 0.537±0.029 | 0.567±0.029 | 0.578±0.018 | 0.584±0.018● | 0.583±0.020● | 0.584±0.016● |
| Comparing | LYN10 | | | | | |
| approach | $\rho = 0.05$ | $\rho = 0.1$ | $\rho = 0.15$ | $\rho = 0.3$ | $\rho = 0.5$ | $\rho = 0.7$ |
| PARM | 0.544±0.020 | 0.611±0.014 | 0.629±0.012 | 0.652±0.012 | 0.661±0.010 | 0.665±0.009 |
| SSPL | 0.586±0.017○ | 0.612±0.013 | 0.624±0.013 | 0.645±0.012 | 0.658±0.014 | 0.669±0.012 |
| PL-KNN | 0.451±0.019● | 0.500±0.008● | 0.509±0.010● | 0.546±0.009● | 0.562±0.012● | 0.581±0.010● |
| CLPL | 0.504±0.027● | 0.581±0.009● | 0.597±0.009● | 0.623±0.012● | 0.631±0.014● | 0.632±0.014● |
| PL-SVM | 0.509±0.015● | 0.571±0.011● | 0.596±0.012● | 0.620±0.010● | 0.629±0.011● | 0.631±0.009● |
| PL-AGGD | 0.572±0.020○ | 0.620±0.010○ | 0.634±0.008 | 0.652±0.009 | 0.655±0.009● | 0.658±0.009● |
| Comparing | LYN20 | | | | | |
| approach | $\rho = 0.05$ | $\rho = 0.1$ | $\rho = 0.15$ | $\rho = 0.3$ | $\rho = 0.5$ | $\rho = 0.7$ |
| PARM | 0.516±0.016 | 0.590±0.013 | 0.607±0.010 | 0.636±0.009 | 0.649±0.010 | 0.653±0.006 |
| SSPL | 0.572±0.012○ | 0.595±0.014 | 0.607±0.014 | 0.637±0.008 | 0.651±0.011 | 0.662±0.013○ |
| PL-KNN | 0.427±0.010● | 0.464±0.012● | 0.490±0.012● | 0.528±0.010● | 0.553±0.011● | 0.568±0.008● |
| CLPL | 0.495±0.018● | 0.560±0.013● | 0.578±0.016● | 0.599±0.008● | 0.610±0.010● | 0.611±0.011● |
| PL-SVM | 0.476±0.030● | 0.546±0.018● | 0.567±0.016● | 0.596±0.008● | 0.608±0.010● | 0.606±0.010● |
| PL-AGGD | 0.546±0.021○ | 0.583±0.008 | 0.594±0.013● | 0.613±0.010● | 0.618±0.010● | 0.621±0.009● |

$0.7, T = 100$]; 2) PL-KNN [14]: An instance-based partial label learning approach which works by $k$NN weighted voting [suggested configuration: $k = 10$]; 3) CLPL [8]: A convex partial label learning approach which works by averaging-based disambiguation [suggested configuration: SVM with squared hinge loss]; 4) PL-SVM [22]: A maximum margin partial label learning approach which works by identification-based disambiguation [suggested configuration: regularization parameter pool with $\{10^{-3}, \cdots, 10^3\}$]; 5) PL-AGGD [26]: A transformation-based partial label learning approach which works by manifold regularization [suggested configuration: $k = 10, \lambda = 1, \mu = 1, \gamma = 0.05$].

Table 2 summarizes characteristics of the experimental data sets used in this paper. Following the widely-used experimental protocol in partial label learning [6, 7, 8, 11], synthetic PL data sets are generated from multi-class UCI data sets with controlling parameter $r$. Here, for any multi-class example $(\boldsymbol{x}_i, y_i)$, one synthetic PL example $(\boldsymbol{x}_i, S_i)$ is generated by randomly adding $r$ labels $\Delta_r \subseteq \mathcal{Y} \setminus \{y_i\}$ into $S_i$, i.e. $S_i = \Delta_r \bigcup \{y_i\}$.[3] Furthermore, five real-world PL data sets from different task domains have also been employed for experimental studies, including Lost [8], LYN10, LYN20 [12] for *automatic face naming*, Mirflickr[13] for *web image classification*, and BirdSong [4] for *bird song classification*.

Table 4: Win/tie/loss counts (pairwise $t$-test at 0.05 significance level) between PARM and each comparing approach on synthetic as well as real-world partial label data sets. [**Controlled UCI data sets**: 36 cases (4 data sets× 9 configurations of $\rho$) for $r = 1, 2$; 27 cases (3 data sets× 9 configurations of $\rho$) for $r = 3$. **Real-world data sets**: 45 cases (5 data sets× 9 configurations of $\rho$)]

| | PARM against | | | | |
| --- | --- | --- | --- | --- | --- |
| | SSPL | PL-KNN | CLPL | PL-SVM | PL-AGGD |
| Controlled UCI data sets ($r = 1$) | 14/21/1 | 26/9/1 | 11/16/9 | 20/14/2 | 10/19/7 |
| Controlled UCI data sets ($r = 2$) | 17/15/4 | 23/13/0 | 10/17/9 | 19/14/3 | 9/22/5 |
| Controlled UCI data sets ($r = 3$) | 18/7/2 | 23/4/0 | 12/14/1 | 18/7/2 | 9/16/2 |
| Real-world data sets | 20/21/4 | 45/0/0 | 31/14/0 | 34/11/0 | 24/16/5 |
| **In Total** | **69/64/11** | **117/26/1** | **64/61/19** | **91/46/7** | **52/73/19** |

On each data set, ten-fold cross validation is performed whose mean accuracy as well as standard deviation are recorded for all comparing approaches. Given the training set $\mathcal{D}_{train}$ and test set $\mathcal{D}_{test}$, a proportion $\rho \in (0, 1)$ of training examples in $\mathcal{D}_{train}$ are sampled to form $\mathcal{D}_P$ and the rest training examples are used to form $\mathcal{D}_U$ by discarding their candidate labeling information. For thorough performance evaluation, we consider varying proportions of PL training examples in this paper with $\rho \in \{0.05, 0.1, 0.15, 0.2, 0.3, 0.4, 0.5, 0.6, 0.7\}$. For semi-supervised comparing approaches PARM and SSPL which learn from PL training examples and unlabeled data, the classification model is trained in $\mathcal{D}_P \bigcup \mathcal{D}_U$ and evaluated on $\mathcal{D}_{test}$. For the other four comparing approaches PL-KNN, CLPL, PL-SVM and PL-AGGD which learn from PL training examples, the classification model is trained in $\mathcal{D}_P$ and evaluated on $\mathcal{D}_{test}$.

As shown in Table 1, the regularization parameters $\lambda$ and $\mu$ for PARM are chosen among $\{0.001, 0.005, 0.01, 0.05, 0.1, 0.5, 1, 5, 10\}$ via cross-validation on training set and $\gamma = 0.01$.

## 4.2 Experimental Results

Due to page limit, Figure 1 and Table 3 report the experimental results on synthetic as well as real-world partial label data sets under certain experimental configurations. Specifically, Figure 1 illustrates how the classification accuracy of each comparing approach changes as $\rho$ (proportion of PL training examples) increases on the synthetic data set `Satimage` (with $r = 1, 2, 3$). In addition, Table 3 gives the classification accuracy of each comparing approach on the real-world partial label data sets (with $\rho \in \{0.05, 0.1, 0.15, 0.3, 0.5, 0.7\}$). Furthermore, Table 4 summarizes the win/tie/loss counts between PARM and each comparing approach (pairwise $t$-test at 0.05 significance level) across all experimental configurations.

Table 4 reports the win/tie/loss counts between PARM and each comparing algorithm based on pairwise $t$-test at 0.05 significance level. As shown in the reported results, we can observe that: a) On synthetic data sets, PARM achieves superior or at least comparable performance to SSPL, PL-KNN, CLPL, PL-SVM and PL-AGGD in 92.9%, 99.0%, 80.8%, 92.9% and 85.9% cases respectively; b) On real-world data sets, compared to the semi-supervised partial label learning approach SSPL, PARM achieves superior performance in 44.4% cases and inferior performance in only 8.9% cases; c) On real-world data sets, compared to partial label learning approaches under supervised setting, PARM significantly outperforms PL-KNN in all cases. Furthermore, PARM significantly outperforms CLPL, PL-SVM and PL-AGGD in 68.9%, 75.6% and 53.3% cases respectively, and has been outperformed by CLPL and PL-SVM in none cases; d) As shown in Figure 1, the performance advantage of PARM over comparing approaches is more pronounced under the challenging cases where $\rho$ (i.e. proportion of PL training examples) is small.

Figure 2 gives the parameter sensitivity analysis for PARM on `BirdSong` data set ($\rho = 0.5$). As shown in Figure 2(a), the performance of PARM is somewhat sensitive w.r.t. $\lambda$ and $\mu$, whose values are chosen via cross-validation on the training set in this paper. As shown in Figure 2(b)-(c), the performance of PARM is relatively stable w.r.t. $\gamma$, whose value is fixed to be 0.01 in this paper.

Figure 3 illustrates how the classification model (i.e. $\|\boldsymbol{w}^{(t)} - \boldsymbol{w}^{(t-1)}\|_2$) and the confidence matrix over unlabeled examples (i.e. $\|\mathbf{F}_U^{(t)} - \mathbf{F}_U^{(t-1)}\|_F$) converge as the number of optimization iterations $t$

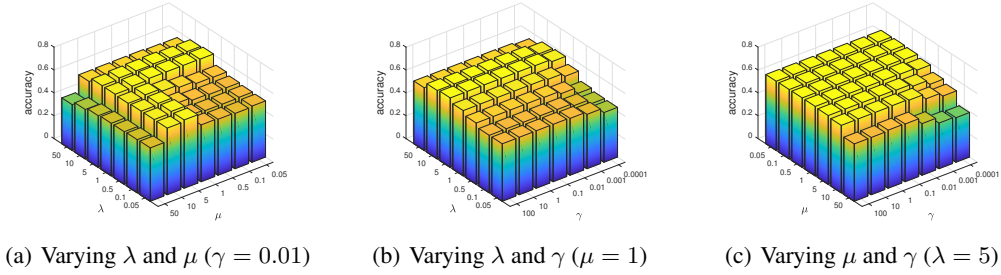

(a) Varying $\lambda$ and $\mu$ ($\gamma = 0.01$)    (b) Varying $\lambda$ and $\gamma$ ($\mu = 1$)    (c) Varying $\mu$ and $\gamma$ ($\lambda = 5$)

Figure 2: Parameter sensitivity analysis for PARM (Data set: BirdSong; $\rho = 0.5$).

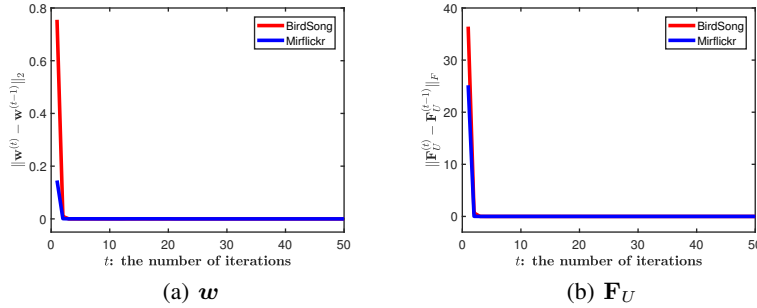

(a) $\boldsymbol{w}$    (b) $\mathbf{F}_U$

Figure 3: Convergence curves of $\boldsymbol{w}$ and $\mathbf{F}_U$ (on BirdSong and Mirflickr).

increases. We can see that the classification model and labeling confidence of unlabeled data converge fast with increasing number of iterations.

## 5 Conclusion

In this paper, the problem of semi-supervised partial label learning is investigated. To learn from both PL training examples and unlabeled data, we introduce confidence-rated margin maximization to jointly optimize predictive model and estimate latent labeling confidence. Comprehensive experiments show that the proposed approach performs favorably against state-of-the-art approaches.

In the future, it would be interesting to investigate ways of enabling the proposed approach to deal with large-scale data sets. Furthermore, other than adopting label propagation to instantiate the labeling confidence of PL examples, it is desirable to explore alternative ways of exploiting the supervision information of PL examples to facilitate model training.

## Broader Impact

In this paper, we study the problem of semi-supervised partial label learning which has been less investigated in weakly supervised learning. The developed techniques can be applied to scenarios where the supervision information collected from the environment is accurate. For ethical use of the proposed approach, one should expect proper acquisition of the candidate labeling information (e.g. crowdsourcing) as well as the unlabeled data. We believe that developing such techniques is important to meet the increasing needs of learning from weak supervision in many real-world applications.

## Acknowledgements

The authors wish to thank the anonymous reviewers for their helpful comments and suggestions. This work was supported by the National Key R&D Program of China (2018YFB1004300), the National

Science Foundation of China (61573104), the China University S&T Innovation Plan Guided by the Ministry of Education, and partially supported by the Collaborative Innovation Center of Novel Software Technology and Industrialization. We thank the Big Data Center of Southeast University for providing the facility support on the numerical calculations in this paper.

## Footnotes

[2]In this paper, $\sigma$, $k$ and $\alpha$ are fixed to be 1, 8 and 0.95 respectively.

[3]For vehicle, the setting $r = 3$ is not considered as there are only four class labels in the label space.

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
