[Supplementary Material]

# Supplement to Semi-Supervised Partial Label Learning via Confidence-Rated Margin Maximization

**Wei Wang    Min-Ling Zhang**[*]

School of Computer Science and Engineering, Southeast University, Nanjing 210096, China
Key Laboratory of Computer Network and Information Integration
(Southeast University), Ministry of Education, China
{wang_w, zhangml}@seu.edu.cn

## 1  Derivation of Eq.(10)

By setting the gradient of $\mathcal{L}\left(\boldsymbol{w}, \boldsymbol{\Xi}, \boldsymbol{\alpha}\right)$ w.r.t. $\boldsymbol{w}$ and $\boldsymbol{\Xi}$ to zero, we can get the following KKT conditions:

$$\frac{\partial \mathcal{L}}{\partial \xi_{il}} = 0 \Rightarrow \begin{cases} \sum_{r=1}^{q} \alpha_{lr}^{i} = \dfrac{\lambda}{p} f_{il} & 1 \leq i \leq p \\[2ex] \sum_{r=1}^{q} \alpha_{lr}^{i} = \dfrac{\mu}{u} f_{il} & p+1 \leq i \leq p+u \end{cases}$$

$$\frac{\partial \mathcal{L}}{\partial \boldsymbol{w}_r} = 0 \Rightarrow \boldsymbol{w}_r = \sum_{i=1}^{p+u}(\sum_{l=1}^{q}\alpha_{rl}^{i} - \sum_{l=1}^{q}\alpha_{lr}^{i})\boldsymbol{x}_i$$

By plugging the first KKT condition into Eq.(9), all the slack variables are counteracted, and the Lagrangian becomes:

$$\begin{aligned}
\mathcal{L}\left(\boldsymbol{w}, \boldsymbol{\Xi}, \boldsymbol{\alpha}\right) =& \frac{1}{2}\|\boldsymbol{w}\|_2^2 + \sum_{i=1}^{p+u}\sum_{l=1}^{q}\sum_{r=1}^{q}\alpha_{lr}^{i}\left(\boldsymbol{w}^{\mathrm{T}}\Phi(\boldsymbol{x}_i, y_r) - \boldsymbol{w}^{\mathrm{T}}\Phi(\boldsymbol{x}_i, y_l) - \delta_{l,r} + 1\right) \\
=& \underbrace{\frac{1}{2}\|\boldsymbol{w}\|_2^2}_{S1} + \underbrace{\sum_{i=1}^{p+u}\sum_{l=1}^{q}\sum_{r=1}^{q}\alpha_{lr}^{i}\boldsymbol{w}^{\mathrm{T}}\Phi(\boldsymbol{x}_i, y_r)}_{S2} \\
&- \underbrace{\sum_{i=1}^{p+u}\sum_{l=1}^{q}\sum_{r=1}^{q}\alpha_{lr}^{i}\boldsymbol{w}^{\mathrm{T}}\Phi(\boldsymbol{x}_i, y_l)}_{S3} + \underbrace{\sum_{i=1}^{p+u}\sum_{l=1}^{q}\sum_{r=1}^{q}\alpha_{lr}^{i}(1 - \delta_{l,r})}_{S4}
\end{aligned}$$

Afterwards, by plugging the second KKT condition into the above equation, we can get:

$$\begin{aligned}
S1 =& \frac{1}{2}\sum_{l=1}^{q}\|\boldsymbol{w}_l\|_2^2 \\
=& \frac{1}{2}\sum_{l=1}^{q}\sum_{i=1}^{p+u}\sum_{j=1}^{p+u}\boldsymbol{x}_i^{\mathrm{T}}\boldsymbol{x}_j(\sum_{r=1}^{q}\alpha_{lr}^{i} - \sum_{r=1}^{q}\alpha_{rl}^{i})(\sum_{r=1}^{q}\alpha_{lr}^{j} - \sum_{r=1}^{q}\alpha_{rl}^{j}) \\
=& \frac{1}{2}\sum_{i=1}^{p+u}\sum_{j=1}^{p+u}\boldsymbol{x}_i^{\mathrm{T}}\boldsymbol{x}_j\sum_{l=1}^{q}(\sum_{r=1}^{q}\alpha_{lr}^{i} - \sum_{r=1}^{q}\alpha_{rl}^{i})(\sum_{r=1}^{q}\alpha_{lr}^{j} - \sum_{r=1}^{q}\alpha_{rl}^{j})
\end{aligned}$$

---

[*]corresponding author

$$S2 = \sum_{i=1}^{p+u}\sum_{l=1}^{q}\sum_{r=1}^{q}\alpha_{lr}^i \sum_{j=1}^{p+u}(\sum_{l=1}^{q}\alpha_{rl}^j - \sum_{l=1}^{q}\alpha_{lr}^j)\boldsymbol{x}_i^{\mathrm{T}}\boldsymbol{x}_j$$

$$= \sum_{i=1}^{p+u}\sum_{j=1}^{p+u}\boldsymbol{x}_i^{\mathrm{T}}\boldsymbol{x}_j \sum_{r=1}^{q}(\sum_{l=1}^{q}\alpha_{rl}^j - \sum_{l=1}^{q}\alpha_{lr}^j)\sum_{l=1}^{q}\alpha_{lr}^i$$

$$= \sum_{i=1}^{p+u}\sum_{j=1}^{p+u}\boldsymbol{x}_i^{\mathrm{T}}\boldsymbol{x}_j \sum_{l=1}^{q}(\sum_{r=1}^{q}\alpha_{lr}^j - \sum_{r=1}^{q}\alpha_{rl}^j)\sum_{r=1}^{q}\alpha_{rl}^i$$

$$S3 = \sum_{i=1}^{p+u}\sum_{l=1}^{q}\sum_{r=1}^{q}\alpha_{lr}^i \sum_{j=1}^{p+u}(\sum_{r=1}^{q}\alpha_{lr}^j - \sum_{r=1}^{q}\alpha_{rl}^j)\boldsymbol{x}_i^{\mathrm{T}}\boldsymbol{x}_j$$

$$= \sum_{i=1}^{p+u}\sum_{j=1}^{p+u}\boldsymbol{x}_i^{\mathrm{T}}\boldsymbol{x}_j \sum_{l=1}^{q}(\sum_{r=1}^{q}\alpha_{lr}^j - \sum_{r=1}^{q}\alpha_{rl}^j)\sum_{r=1}^{q}\alpha_{lr}^i$$

$$S4 = \sum_{i=1}^{p+u}\sum_{l=1}^{q}\sum_{r=1}^{q}\alpha_{lr}^i - \sum_{i=1}^{p+u}\sum_{l=1}^{q}\sum_{r=1}^{q}\alpha_{lr}^i\delta_{l,r}$$

$$= \sum_{i=1}^{p}\sum_{l=1}^{q}\frac{\lambda}{p}f_{il} + \sum_{i=p+1}^{p+u}\sum_{l=1}^{q}\frac{\mu}{u}f_{il} - \sum_{i=1}^{p+u}\sum_{l=1}^{q}\sum_{r=1}^{q}\alpha_{lr}^i\delta_{l,r}$$

$$= -\sum_{i=1}^{p+u}\sum_{l=1}^{q}\sum_{r=1}^{q}\alpha_{lr}^i\delta_{l,r} + \lambda + \mu$$

Therefore,

$$S2 - S3 = \sum_{i=1}^{p+u}\sum_{j=1}^{p+u}\boldsymbol{x}_i^{\mathrm{T}}\boldsymbol{x}_j \sum_{l=1}^{q}(\sum_{r=1}^{q}\alpha_{lr}^j - \sum_{r=1}^{q}\alpha_{rl}^j)(\sum_{r=1}^{q}\alpha_{rl}^i - \sum_{r=1}^{q}\alpha_{lr}^i)$$

$$= -\sum_{i=1}^{p+u}\sum_{j=1}^{p+u}\boldsymbol{x}_i^{\mathrm{T}}\boldsymbol{x}_j \sum_{l=1}^{q}(\sum_{r=1}^{q}\alpha_{lr}^j - \sum_{r=1}^{q}\alpha_{rl}^j)(\sum_{r=1}^{q}\alpha_{lr}^i - \sum_{r=1}^{q}\alpha_{rl}^i)$$

By rearranging the above equations, the dual problem, i.e. $\max_{\boldsymbol{\alpha}} \min_{\boldsymbol{w},\boldsymbol{\Xi}} \mathcal{L}(\boldsymbol{w},\boldsymbol{\Xi},\boldsymbol{\alpha})$, can be equivalently formulated as:

$$\min_{\boldsymbol{\alpha}} \quad \frac{1}{2}\sum_{i=1}^{p+u}\sum_{j=1}^{p+u}\boldsymbol{x}_i^{\mathrm{T}}\boldsymbol{x}_j \sum_{l=1}^{q}\left(\sum_{r=1}^{q}\alpha_{lr}^i - \sum_{r=1}^{q}\alpha_{rl}^i\right)\left(\sum_{r=1}^{q}\alpha_{lr}^j - \sum_{r=1}^{q}\alpha_{rl}^j\right) + \sum_{i=1}^{p+u}\sum_{l=1}^{q}\sum_{r=1}^{q}\alpha_{lr}^i\delta_{l,r}$$

$$\text{s.t. } \alpha_{lr}^i \geq 0, \quad (1 \leq i \leq p+u,\ 1 \leq l,r \leq q)$$

## 2 Derivation of Eq.(12)

Without loss of generality, by fixing the values of other Lagrangian multipliers in $\boldsymbol{\alpha}$, the $i$-th sub-QP problem can be formulated as follows:

$$\min_{\boldsymbol{\alpha}^i} \quad \frac{1}{2}\boldsymbol{x}_i^{\mathrm{T}}\boldsymbol{x}_i \sum_{l=1}^{q}\left(\sum_{r=1}^{q}\alpha_{lr}^i - \sum_{r=1}^{q}\alpha_{rl}^i\right)\left(\sum_{r=1}^{q}\alpha_{lr}^i - \sum_{r=1}^{q}\alpha_{rl}^i\right)$$

$$+ \sum_{j\neq i}\boldsymbol{x}_i^{\mathrm{T}}\boldsymbol{x}_j \sum_{l=1}^{q}\left(\sum_{r=1}^{q}\alpha_{lr}^i - \sum_{r=1}^{q}\alpha_{rl}^i\right)\left(\sum_{r=1}^{q}\alpha_{lr}^j - \sum_{r=1}^{q}\alpha_{rl}^j\right) + \sum_{l=1}^{q}\sum_{r=1}^{q}\alpha_{lr}^i\delta_{l,r}$$

$$\text{s.t. } \alpha_{lr}^i \geq 0, \quad (1 \leq l,r \leq q)$$

By introducing $\mathbf{M}$, $\mathbf{N}$ and $\mathbf{C}$,

$$\sum_{l=1}^{q}\left(\sum_{r=1}^{q}\alpha_{lr}^{i}-\sum_{r=1}^{q}\alpha_{rl}^{i}\right)\left(\sum_{r=1}^{q}\alpha_{lr}^{j}-\sum_{r=1}^{q}\alpha_{rl}^{j}\right)$$
$$=(\mathbf{M}\mathrm{vec}(\boldsymbol{\alpha}^{i})-\mathbf{N}\mathrm{vec}(\boldsymbol{\alpha}^{i}))^{\mathrm{T}}(\mathbf{M}\mathrm{vec}(\boldsymbol{\alpha}^{j})-\mathbf{N}\mathrm{vec}(\boldsymbol{\alpha}^{j}))$$
$$=(\mathbf{C}\mathrm{vec}(\boldsymbol{\alpha}^{i}))^{\mathrm{T}}(\mathbf{C}\mathrm{vec}(\boldsymbol{\alpha}^{j}))$$

Therefore, Eq.(12) can be formulated as

$$\min_{\boldsymbol{\alpha}^{i}}\ \frac{1}{2}\boldsymbol{x}_{i}^{\mathrm{T}}\boldsymbol{x}_{i}\mathrm{vec}(\boldsymbol{\alpha}^{i})^{\mathrm{T}}\mathbf{C}^{\mathrm{T}}\mathbf{C}\mathrm{vec}(\boldsymbol{\alpha}^{i})+\left(\sum_{j\neq i}\boldsymbol{x}_{i}^{\mathrm{T}}\boldsymbol{x}_{j}\mathbf{C}^{\mathrm{T}}\mathbf{C}\mathrm{vec}(\boldsymbol{\alpha}^{j})+\mathrm{vec}(\mathbf{I}_{q\times q})\right)^{\mathrm{T}}\mathrm{vec}(\boldsymbol{\alpha}^{i})$$

$$\text{s.t. } \alpha_{lr}^{i}\geq 0, \quad (1\leq l,r\leq q)$$