[Reviews · NeurIPS 2020]

Review 1

Summary and Contributions: The authors propose a methodology for labeling unseen instances using weakly supervised data. The weak supervised data is given in terms of unlabeled data an data labeled with candidate set of labels. They propose an optimization problem for obtaining the labels of the data, and the problem is solver by iterating a set of quadratic programming sub-problems until convergence.

Strengths: -The work proposes an alternative to SSPL [22] for labeling unlabeled data and data labeled using candidate sets. -The proposal seems to outperform SSPL. -The problem is formally formulated as an optimization problem that is solved by an iterative procedure. The problem and the iterative solution seems to be correct.

Weaknesses: -The literature review should be more focused in describing in detail the works that are more related, e.g. [22], rather than provide list of references without further details. -In the experiments the candidate sets are small (at most of size 4). Thus, it is not clear to me the effect of the size of the candidate sets on the performance of the proposal. -In the experiments the candidate sets are of a fixed size. It should be consider to illustrate the effect of candidate sets of different sizes.

Correctness: -The claims and the proposed methodology seems to be correct -The empirical analysis is also correct, however, I miss experimental results with larger sizes and variable sizes for the candidate.

Clarity: - Yes but sometimes it becomes hard to follow due to the technical content. It could be improved by including some intuitions about why PARM works

Relation to Prior Work: - More details about the closest approaches could improve the relation of the proposal with the state of the art.

Reproducibility: Yes

Additional Feedback: - I can not find the definition of sigma (lines 97 and 105) - I think that for improving the visual comparison the Y axis of Figure 1 should be in the same scale in all subfigures


Review 2

Summary and Contributions: This paper studies an interesting problem setting called semi-supervised partial label learning. To solve this problem, this paper adopts a two-stage method. For the first stage, label propagation is used to produce labeling confidence for partial label examples. For the second stage, a maximum margin formulation is introduced to jointly enable the induction of the predictive model and the estimation of labeling confidence over unlabeled data. Experiments have validated the effectiveness of the proposed method. I feel that although this paper presents a reasonable solution to semi-supervised partial label learning, the novelty is limited. ====================================After Author Rebuttal=========================================== I have read other reviews and the author response, which fully addressed my questions. For the novelty, I have re-checked this paper and compared with SSPL [22]. SSPL is the first method for semi-supervised partial label learning, but SSPL does not have the inductive ability (it needs to further use kNN to obtain predictions on unseen examples). This paper addresses this issue by proposing a decent and reasonable maximum margin formulation. This formulation is an improvement over [17], but this paper focuses on a more challenging problem, i.e., semi-supervised partial label learning while [17] is only for partial label learning. This paper deals with unlabeled data with manifold regularization and improves the model training by an additional confidence weighting strategy. For semi-supervised partial label learning, I consider that this paper would be an important paper in this area and I am convinced that the novelty is enough. For the convergence of the proposed method, the authors have provided the convergence analysis in the rebuttal. I am satisfied with that. So I would like to increase my score and give acceptance to this paper.

Strengths: 1. The solution is intuitive and reasonable. 2. Empirical results have validated the effectiveness of the proposed method.

Weaknesses: I think the major problem of this paper is that the novelty is limited. It seems that nearly all the components in the proposed method have been used, and the combination manner is also not so novel. 1. The two-stage method is widely used to deal with partial label examples. The first stage in this paper adopts the widely used label probagation strategy to obtain labeling confidence for partial label examples. However, SSPL [22] also adopts this strategy (which obtain labeling confidence for both partial label examples and unlabeled examples). So there seems no novelty for the first stage. 2. For the second stage, a modified maximum margin formulation is introduced, which can jointly enable the induction the induction of predictive model and the estimation of labeling confidence over unlabeled data. The formulation of the second stage mainly follows [17]. There are two differences between this paper and [17]: Firstly, this paper uses the obtained labeling confidence of partial examples to give different weights on the losses of different examples. Secondly, the unlabeled data is used in the formulation by using the widely-used manifold regularization, so that the labeling confidence of unlabeled data will be jointly estimated. Generally, I think the improvement over [17] is intuitive and reasonable. But I think that the novelty is not enough, especially on such a prestiguous venue NeurIPS. 3. The alternating optimization is a common optimization solution, which has also been adopted by a related paper [21]. But unlike [21], this paper does not provide any theoretical analysis or empirical evidence about the convergence of the modified maximum margin formulation in the second stage. 4. This paper may not effectively deal with large-scale datasets because it uses alternating optimization and needs to construct a similarity graph in advance.

Correctness: Yes.

Clarity: This paper is well-written in general.

Relation to Prior Work: Yes.

Reproducibility: Yes

Additional Feedback: Convergence analysis of the modified maximum margin formulation in the second stage could be discussed. There is a minor issue: In Abstract and Introduction, this paper fails to clearly state that the proposed method is two-stage. Instead, this paper seems to only describe the design of the second-stage (a modified maximum margin formulation with unlabeled data). I suggest the authors to further improve that for a better presentation.


Review 3

Summary and Contributions: This paper studies the problem of semi-supervised partial label learning, which is an interesting weakly supervised learning problem where unlabeled data are utilized to induce predictive model with partial label examples. Accordingly, a first attempt to semi-supervised partial label learning based on maximum margin formulation is proposed. The performance advantage of the proposed approach over state-of-the-art approaches are clearly validated via extensive experimental studies.

Strengths: 1. The problem studies in this paper, i.e. semi-supervised partial label learning, is interesting and stands as an important topic for weakly supervised learning. 2. The maximum margin formulation developed in this paper is well motivated and clearly presented. 3. Comprehensive experiments are performed on synthetic as well as real-world data sets to show the effectiveness of the proposed approach.

Weaknesses: 1. In this paper, the labeling confidences over partial label examples (F_P) are estimated via label propagation and kept unchanged in the follow-up optimization procedure. Is it possible to jointly optimize F_P with the predictive model as in Eq.(4)? 2. It is impressive that the proposed approach achieves significantly better performance over state-of-the-art comparing approaches. Furthermore, it would be more informative if some fine-grained conclusions can be drawn w.r.t. the properties of the data sets. For instance, which factors of the data sets would have stronger influence on the performance of the proposed approach?

Correctness: Yes.

Clarity: The whole paper is well written and easy to follow.

Relation to Prior Work: Yes

Reproducibility: Yes

Additional Feedback: Please refer to the comments given in the "Weaknesses" section. I have read the author response and peer's reviews. The author response is fine for me.


Review 4

Summary and Contributions: This paper studies the semi-supervised partial label learning problem where there is a set of candidate labels (only one valid) for each training point. They propose a maximum margin formulation, specifically, confidence-rated margin is maximized by preserving labeling confidence manifold structure between partial label training examples and and unlabeled examples.

Strengths: The strength of the paper lies in the alternating optimization procedure they use to update the predictive model and labeling confidence.

Weaknesses: Sample complexity bounds (both for labels and unlabeked examples) are lacking.

Correctness: The experiments are comprehensive and the methodology is correct. The convergence rate curve of the classification model is interesting; but it would be desirable to have more experiments on a variety of larget datasets.

Clarity: The paper is relatively well written.

Relation to Prior Work: Related work is not broad enough to cover the vast literature of semi-supervised learning.

Reproducibility: Yes

Additional Feedback:

[Author Response · NeurIPS 2020]

First of all, we wish to sincerely thank the anonymous reviewers for their time and efforts in reviewing our NeurIPS
submission #5474. Next, we would like to provide responses to major concerns raised in the reviewing comments:

**[Limited novelty]**
In this paper, the first maximum margin solution towards the problem of semi-supervised partial label learning is
proposed. To the best of our knowledge, the SSPL [22] approach corresponds to the only prior work on the same
problem studied in this paper. The key differences between SSPL and the proposed PARM approach correspond to:
1) SSPL employs graph-based label propagation for estimating the labeling confidence over both partial label and
unlabeled examples, while PARM employs label propagation to instantiate the labeling confidences over partial label
examples. The labeling confidences over unlabeled examples are estimated by PARM based on follow-up maximum
margin procedure; 2) Due to the transductive nature of graph-based methods, SSPL is not meant to be able to make
predictions on unseen examples during testing phase. As a remedy, SSPL further applies $k$NN rule over training
examples with estimated labels to enable inductive prediction on unseen examples. Due to the inductive nature of
maximum margin approach, PARM is capable of making predictions on unseen examples without resorting to extra
procedure. In the revised version, we will make this clearer in the "Related Work" section.

**[Variable sizes of candidate label set]**
To illustrate the performance of PARM on datasets with larger and variable size of candidate label set, we enlarge
the candidate label set of partial label examples in `Lost` and `BirdSong` datasets by randomly adding irrelevant labels
into their candidate label set. Consequently, by increasing the proportion ($\rho$) of partial label examples with randomly
added irrelevant labels, the size of candidate label set would vary from 8 to 10 for `Lost` dataset and from 5 to 9 for
`BirdSong` dataset respectively. Figure 1 illustrates how PARM and the comparing approaches perform as $\rho$ increases
from 0.05 to 0.7. The results clearly show the advantage of PARM in learning from partial label examples with larger
and variable size of candidate label set.

(a) On `Lost` data set        (b) On `BirdSong` data set

Figure 1: Classification accuracy of PARM and each comparing approach with varying size of candidate label set.

**[Convergence analysis]**
Figure 2 illustrates how the classification model (i.e. $\|\boldsymbol{w}^{(t)} - \boldsymbol{w}^{(t-1)}\|_2$) and the confidence matrix over unlabeled
examples (i.e. $\|\mathbf{F}_U^{(t)} - \mathbf{F}_U^{(t-1)}\|_F$) converge as the number of optimization iterations $t$ increases. The high convergence
rate of PARM is desirable for dealing with data sets with larger scale.

(a) $\boldsymbol{w}$        (b) $\mathbf{F}_U$

Figure 2: Convergence curves of $\boldsymbol{w}$ and $\mathbf{F}_U$ (on `BirdSong` and `Mirflickr`).

**[Definition of $\sigma$]**
The parameter $\sigma$ corresponds to the width of Gaussian kernel, which is fixed to be 1 in this paper (pp.3, footnote 1).

[Meta-Review · NeurIPS 2020]

The paper proposes new a method for learning under partially determined labels. The paper is well written and empirical results are convincing.